# Organometallic Coatings for Electroluminescence Applications

**Silviu Polosan, Iulia Corina Ciobotaru and Claudiu Constantin Ciobotaru ***

National Institute of Materials Physics, Magurele 077125, Romania; silv@infim.ro (S.P.);
corina.ciobotaru@infim.ro (I.C.C.)
***** Correspondence: claudiu.ciobotaru@infim.ro; Tel.: +40-2148-268

**Abstract:** Organometallic compounds embedded in thin films are widely used for Organic Light-Emitting Diodes (OLED), but their functionalities are strongly correlated with the intrinsic properties of those films. Controlling the concentration of the organometallics in the active layers influences the OLED performances through the aggregation processes. These aggregations could lead to crystallization processes that significantly modify the efficiency of light emission in the case of electroluminescent devices. For functional devices with organometallic-based thin films, some improvements, such as the optimization of the charge injection, are needed to increase the light output. One dual emitter $IrQ(ppy)_2$ organometallic compound was chosen for the aggregation correlations from a multitude of macromolecular organometallics that exist on the market for OLED applications. The choice of additional layers like conductive polymers or small molecules as host for the active layer may significantly influence the performances of the OLED based on the $IrQ(ppy)_2$ organometallic compound. The use of the CBP small molecule layer may lead to an increase in the electroluminescence versus the applied voltage.

**Keywords:** organometallics; thin films; coatings; OLED; electroluminescence; dual emission

## 1. Introduction

Organometallics are becoming increasingly used for their good photophysical properties and chemical stability, as either nanopowders or thin films. Starting with the early first Organic Light-Emitting Diodes (OLED) in 1987 [1], these compounds cover a wide range of applications, from biology to solar cells. In biology, these organometallics are used in combination with inorganic nanoparticles like silica for in vivo experiments [2]. High-resolution characterization of these organic/inorganic structures reveals the presence of phosphorescent clusters trapped within the diatom organic residual matter. These compounds are used as catalyzers [3,4] and in vitro anticancer activity studies, being a good candidate to anticancer drug cisplatin against A549 cancer cells [5–8]. The main photophysical properties of these organometallics have been used in the field of marking processes of amino acids [9] and, due to their fluorophores' capabilities in imaging, sensing and therapy processes [10]. Other applications of organometallics are in photo redox catalysis, as a part of green synthesis methods [11]; to activate water oxidation and reduction catalysts in solution [12]; for photoactive centers [13] and optical sensors [14]; and in the field of photovoltaic applications [15–20]. All of these applications have as main factors the photophysical capabilities of the organometallic compounds that are strongly influenced by the aggregation processes.

From far, the main application of these organometallics remains the Organic Light-Emitting Diodes (OLED) as dopants in transparent and conductive thin polymers. This technology is already applied in the market as color displays on small devices or even in the larger OLED television, but their time stability remains a major problem. In OLED applications, the organometallic compounds have

been used for their electroluminescent capability as a process in which the charge carrier injection leads to radiative recombination on these organometallic molecules, enabling the light emission. The main target in this field remains the construction of a functional Red–Green–Blue stable matrix that can be obtained either by using three different molecules of organometallic, each with their phosphorescence or by one organometallic molecule with all-three-color phosphorescence. In the first case, the mixing of different organometallics leads to aggregation processes for each type of emitter, resulting in the quenching of the emission. This problem appears also in the case of highly doped active layers that form the OLED structures, as is underlined later in the present paper. In the last case, the mixing of ligands on the same molecule require different physical precautions because the red color is given by a small bandgap where the radiative transitions are competing with the non-radiative one, while for the blue emission, the large bandgap makes difficult the formation of the excitons, after the charges injection in the OLED.

The sandwich structure of OLED is constructed by dispersing the active organometallic compound in transparent and conductive materials (polymers or small molecules). These active layers are then stacked between one hole transport and one electron transport layers and capped with electrodes. Usually, these electrodes are indium thin oxide (ITO) for the positive electrodes, and metals, like aluminum, gold or silver, for the negative electrodes, with good electron injection (Figure 1). In our case, the LiF layer plays an important role in the reduction of the injection barrier at the interface with the active layer, while the electron transport layer is missing due to their dual characteristics of the organometallic structure that acts as ETL and electroluminescent material similar with tris-(8-hydroxyquionline) aluminum (Alq$_3$).

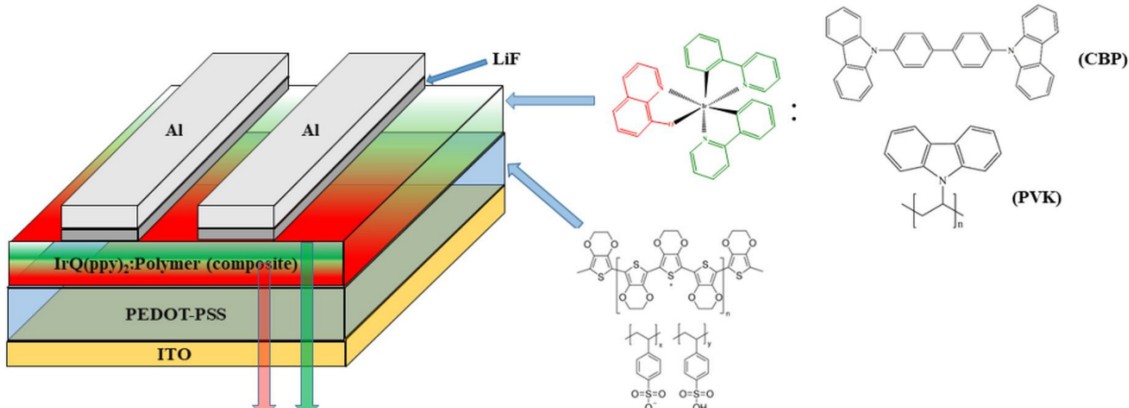

**Figure 1.** OLED sandwich structure based on IrQ(ppy)$_2$ organometallic.

Apart from the single-color organometallics like Alq$_3$ or Ir(ppy)$_3$, the obtaining of multicolor phosphorescence requires a tuning process of their emissions by combining cyclometalates or ancillary ligands. In these cases, the electron transfer from the metal ion to the ligands reveals two or three fundamental colors. The cyclometalates compounds based on Ir(III) ions exhibit higher emission quantum yields of the excited molecules compare with those with ancillary ligands [21]. Among them, special attention was devoted to the organometallics with homo and heteroleptic ligands based on phenylpyridine [22]. Different substitutions have been done as an interplay between cyclometalated and ancillary ligands for fine-tuning of phosphorescence [23]. For blue–green phosphorescence, several combinations of cyclometalated ligands with modified side chains have been proposed, but their phosphorescence is just slightly shifted toward the blue color (around 488 nm) [24], and some questions related to their efficiency in this region remain because the electroluminescence does not vary with the increasing of the applied voltage [25]. As we will see, this fact can be related to the right choice of the conductive polymer.

One of the most difficult phosphorescences of the organometallics is the red color, due to the interplay between the radiative and non-radiative transitions, which could lead to a quenching

mechanism. Several attempts have been made in this field, one of these iridium-based organometallics being bis-cyclometalated iridium complex with a nitrogen-containing ligand, which enhances the phosphorescence quantum yields in the red region of the visible spectrum [26]. This red phosphorescence is connected with the strongly $\pi$-donating process in ancillary ligands. The presence of the ancillary ligands and the nitrogen-containing can lead to larger radiative rates and higher phosphorescence quantum yields for red-emitting complexes. The non-radiative decay rate is inversely related to the energy difference between the ground state and the excited state of the emissive molecules [27], while the radiative rate has a cubic dependence on the transition energy and thus it is expected to be smaller for lower-energy emitters [28].

Another reason can be related to the strength of the spin–orbit coupling between singlet and triplet states, which increases the allowed spin-forbidden radiative transitions by relaxing spin-selection rules [29]. A class of these organometallics with near-infrared emission has been developed, but it is not very useful for the RGB emitters [30]. Besides the organometallics' electroluminescence efficiency, an important factor is given by the charge transport across the OLED sandwich structures. A typical OLED sandwich structure consists of the 4,4(-*N*,*N*)-dicarbazole-biphenyl (CBP) doped with organometallics and was used for efficient green phosphorescent organic light-emitting devices [31]. Several attempts of improving the charge transport across the OLED sandwich structures have been made to increase the external quantum efficiency [32,33].

Some important parameters of the OLED structures can be theoretically simulated by using different quantum chemistry software, like Gaussian or Amsterdam Density Functional, which allow the estimation of the HOMO and LUMO levels for the organometallics, together with their charge transport properties and the influence of the spin–orbit coupling on the photophysical properties of these materials [34–36].

The main purpose of this manuscript is related to the changing of the electroluminescence as a result of the aggregation processes, which can lead to the crystallization of the organometallic. This fact requires different mechanisms of the charge transport, by hopping in the case of diluted (8 wt.%) $IrQ(ppy)_2$ and through the $\pi$–$\pi$ bonding for the partial crystalline sample with 20% wt. $IrQ(ppy)_2$. The variation of the electroluminescence between green and red color was observed for the first time and is now correlated with the previous results, where the photoluminescence and X-ray diffraction were correlated with the crystallization processes, which take place at higher concentrations of organometallic [37]. Some theoretical predictions were given for metal–oxide frameworks materials concerning the electrical conductivity through $\pi$–$\pi$ stacked molecules [38]. Zhang et al. compare the current–voltage curves of $Ir(ppy)_3$ between the neat film and embedded in CBP (7 wt.%) but increasing the thickness of the doped CBP and show that the forward voltage increases gradually with increasing $Ir(ppy)_3$ thickness [39].

The structural properties of $IrQ(ppy)_2$, investigated by X-ray diffraction, show a triclinic structure P-1, with the following parameters $a = 10.3$; $b = 13.0$; $c = 13.8$; $\alpha = 65.5°$; $\beta = 86.7°$; $\gamma = 60°$, while the Scanning Electron Microscopy (SEM) and cathodoluminescence images confirm this structure. More details concerning thermal analysis, X-ray photoelectron spectroscopy and backscattering electron microscopy can be found in the reference [40].

The photophysical properties of $IrQ(ppy)_2$ are connected with the degree of dispersion between the molecules. While in the powder sample, the red color dominates the photoluminescence, centered at 640 nm (1.93 eV), in solution, the green color centered at 521 nm (2.38 eV) is dominant. The aggregation processes induce inter-ligand charge transfer and enhance the red color. Proper dispersion of the $IrQ(ppy)_2$ leads to a balance between the green and red phosphorescence, resulting in an allover orange color. This aspect was demonstrated by photoluminescence spectra of $IrQ(ppy)_2$ dispersed in CBP (8% and 15%), under 408 nm excitation, which is mainly green at low concentrations and red at high concentrations. The same is true for the $IrQ(ppy)_2$ dispersed in dichloromethane between 0.15 and 0.8 mg/mL, but also in PMMA between 2.1 and 14.8 wt.%.

Concerning the lifetime measurements at room temperature, the green phosphorescence gives 0.98 μs in $CH_2Cl_2$ and 1.32 μs in PMMA, while the red one gives 0.67 μs in $CH_2Cl_2$ and 8.98 μs in PMMA under 375 nm excitation. The lifetime measurements were performed by using a Jobin–Yvone spectrophotometer equipped with a 375 nm diode laser with a discharge time ranging from 1 ns up to a few tens of ps [41]. For the absorption spectra recorded on solutions containing the $IrQ(ppy)_2$, the absorption peaks between 300 and 500 nm confirm the metal-to-ligand charge transfer and the absorption peaks of ligands below 300 nm.

In this paper, a dual emitter organometallic compound $IrQ(ppy)_2$ is extensively presented, underlying three main aspects: (a) the effects of the conductive layer and the structure of cathodes on the electroluminescence of this organometallic. Some aspects concerning the concentration of $IrQ(ppy)_2$ in the conductive polymers on the external quantum efficiency of the OLED devices are described, as a part of solution-processed thin films [42]; (b) the $IrQ(ppy)_2$ concentration in the active layer was changed to monitor the OLED performances, considering the doping level, which can influence the charge transport across the sandwich structure and then further the external quantum efficiency [43]; (c) and the effect of the LiF buffer layer between the aluminum cathode and the OLED active layer, to reduce the injection barrier of the electrons. This fact was done for comparison with the previous results, in which the low-level electroluminescence was connected with the direct injection of electrons into the active layer.

## 2. Materials and Methods

All reagents were purchased from Sigma-Aldrich (St. Louis, MO, USA) and used without preliminary purification. The glass/ITO substrates, aluminum and the epoxy resin were taken from Ossila. The synthesis of the $IrQ(ppy)_2$ is described below. The obtained powder was dissolved in chloroform together with polymer or small molecule materials and spin-coated with the Ossila spin-coater. PEDOT:PSS dispersed in water was spin-coated with the same equipment. The LiF buffer layer and aluminum cathodes were deposited with a Bestec thermal evaporator (Bestec, Berlin, Germany). The thicknesses of these layers were evaluated with FR-portable Thetametrisis in the reflection mode, while the metal cathodes were measured with Mitutoyo step-scan profilometer (Mitutoyo, IL, USA).

The current–voltage curves were recorded with a Keithley 2450 source-meter (Cleveland, OH, USA), while the luminance was measured with a Konica Minolta CS-2000 spectroradiometer (Konica Minolta, Tokyo, Japan). The photoluminescence was excited with a 405 nm photodiode with 160 mW electric power.

### 2.1. Synthesis of IrQ(ppy)₂ Organometallic

The common method for synthesis of organometallic compounds, especially based on Ir(III) and Pt(III), follows the Nonoyama mechanisms [44–48]. Briefly, reactions of phenylpiridine with chloride salts of iridium or platinum give some dichloro-bridge dimers of Ir(III) or Pt(III). In the case of single-color phosphorescent material, like $Ir(ppy)_3$, the facial structure of this organometallic compound appears as a side-product to a dichloro-bridge dimer of iridium [49]. Drying procedure on $MgSO_4$ catalyzer leads to *fac*-$Ir(ppy)_3$.

In the case of $IrQ(ppy)_2$, the organometallic is formed by a central Ir(III) ion bonded with two phenylpiridine ligands and one quinoline ligand. This type of organometallic requires a two-step reaction. In the first step, the $IrCl_3*H_2O$ powder and 2.5 equivalent of 2-phenylpiridine ligands were dissolved in a 3:1 mixture of 2-ethoxyethanol and water. This slurry was heated at 100 °C for 24 h. After cooling to room temperature, the precipitate, [CˆN₂Ir(μ-Cl)₂IrCˆN₂] known as the intermediate compound, was filtered and washed with water [50]. The second step is dedicated to adding one new ligand, which replaces the chlorine ions. For the red emission, the substitution of the chlorine ions was done with 8-hydroxyquinoline, using the following procedure: the mixture containing 2-etoxyethanol, [CˆN₂Ir(μ-Cl)₂IrCˆN₂], 8-hydroxyquinoline and sodium carbonate was heated and refluxed under

nitrogen atmosphere for 24 h. It was then cooled to room temperature and was filtered to extract the precipitate from the solution. The crude product was dissolved in dichloromethane and filtered to remove the sodium carbonate catalyst. The IrQ(ppy)$_2$ powder was obtained after crystallization from dichloromethane. The powder was dried for 12 h at 60 °C [50].

### 2.2. Devices Fabrication

In a classical method, the PEDOT:PSS is used as a hole transport layer and deposited by spin-coating method at 5000 rpm for 30 s, on the ITO pre-patented glasses. The spinning procedure was followed by thermal annealing of samples at 120 °C for 10 min. Previously, the glass/ITO substrates were cleaned in Hellmanex 2% *v/v* solution, rinsed in deionized water and washed with acetone and then with isopropyl alcohol, dried and plasma-treated for better adhesion. The active layer, consisting from poly(9-vinylcarbazole) (PVK) polymer or 4,4'-Bis(9-carbazolyl)-1,1'-biphenyl, 4,4-*N*,*N*'-Dicarbazole-1,1'-biphenyl (CBP) small molecule and IrQ(ppy)$_2$ dissolved in chloroform and then spin-coated at 3000 rpm for 30 s. Three samples, with concentrations 8, 10 and 20 wt.% of IrQ(ppy)$_2$/PVK(CBP) were obtained. A buffer layer of LiF of about 0.8 nm was deposited by thermal evaporation from wolfram boats at $I = 1.4$ A, $p = 3 \times 10^{-6}$ mbar, at rates from 0.02 to 0.04 nm/s, obtaining a thickness of 0.8 nm. Over the LiF layer was deposited Al by thermal evaporation from wolfram boats at $I = 1.8–2$ A, $p = 2 \times 10^{-6}$ mbar, $R = 0.01–0.2$ nm/s, obtaining a thickness of 100 nm.

For the metallic cathodes, the thicknesses were evaluated by using a Mitutoyo step-scan profiler (Mitutoyo, Tokio, Japan), which enables thickness measurements starting with 20 nm. The structure of the IrQ(ppy)$_2$-based OLED devices is given in Figure 2a,b.

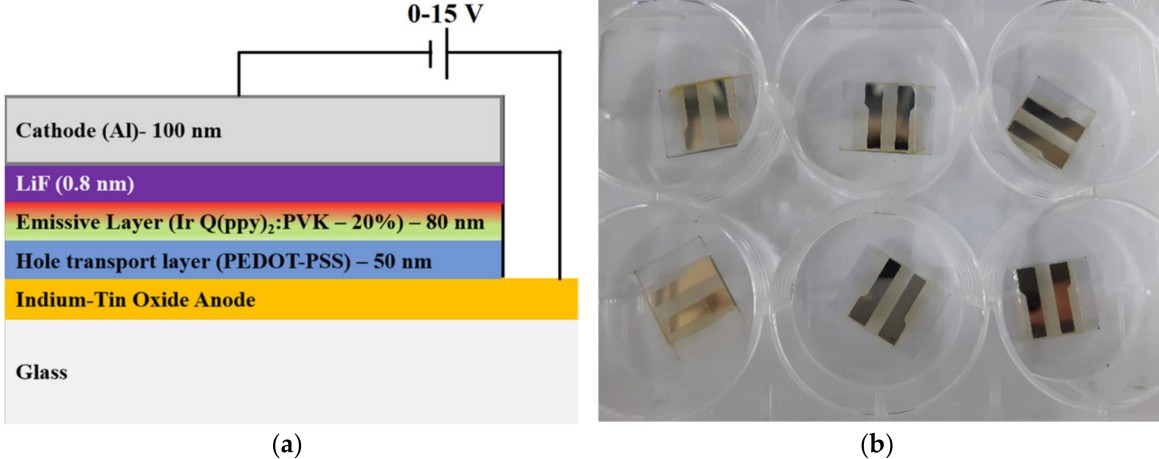

| (a) | (b) |

**Figure 2.** Final structure (**a**) and devices of IrQ(ppy)$_2$-based devices (**b**).

Finally, the obtained sandwich structures were encapsulated over the metal cathodes by using epoxy resin E132 from Ossila (Sheffield, UK), and then they were UV-exposed for 2–3 min, to fix the epoxy resin. Electroluminescence measurements were performed on sealed OLED devices, to avoid the oxygen contamination of these devices. All measurements were performed at room temperature and ambient conditions.

## 3. Results

### 3.1. Current–Voltage Characteristics: Electroluminescence of IrQ(ppy)$_2$ in PVK

The current–voltage characteristics were evaluated with a Keithley 2450 source-meter, in transversal mode, with a linear sweeping voltage from 0 to 15 V, and a step scan of 0.2 V.

The general form of the current density is given by the following formula:

$$j = \frac{9}{8} \varepsilon \mu \frac{V^{m+1}}{d^3} \tag{1}$$

where $\varepsilon = \varepsilon_0 \varepsilon_r$, with $\varepsilon_r$ being the dielectric constant of the polymer, and $\mu$ being the charge mobility; the mobility can be extracted from the fitting curve if this mobility is independent of the temperature. [51]. If the current is controlled by a space-charge mechanism, considering the existence of a single discrete trap energy level, the current density is expressed as follows:

$$j = \frac{9}{8} \varepsilon \mu_{eff} \frac{V^2}{d^3} \tag{2}$$

where $\mu_{eff}$ is the effective charge carrier mobility [52].

The luminance properties of OLED devices are quantified in terms of light emission measured in candela per square meter (cd/m$^2$). Besides current–voltage measurements that allow determination of the current density, the luminance allows calculation of power efficiency in lumens/W and external quantum efficiency of these devices.

Figure 3a reveals the current–voltage curves at different concentrations of IrQ(ppy)$_2$ in PVK polymer, while Figure 3b shows the luminance versus applied bias for the sample with 10% IrQ(ppy)$_2$ in PVK. As one can see, the use of PVK polymer does not change the luminance intensity with the increasing of bias, giving the same orange color as a mixture of green and red electroluminescence. Figure 3c shows the luminance measured at 15 V for the IrQ(ppy)$_2$:PVK (10%) sample and the photoluminescence on the same sample excited at 405 nm. As can be seen, there is an overlap between spectra. In the inlet is given the photoluminescence dependence versus concentration of IrQ(ppy)$_2$ when the samples are excited at 405 nm.

Figure 4 presents the current–voltage curves for the same doping (20% IrQ(ppy)$_2$) in both PVK polymer and CBP small-molecule material. As one can see, there is just a slight increase of the current in CBP at 15 V. The inset shows the current–voltage curves in the logarithmic scales for identification of the conduction mechanisms for each type of conductive host material. The first region with $m = 0$ is specific for the ohmic conductivity up to 3 V. In the second region, with $m = 1$ from 3 to 10 V is specific for the Space Charge Limited Currents (SCLC) where the traps are filled and the free charge carrier gives the measured mobility. Finally, the last zone has $m = 5$ between 10 to 15 V is dominated by the trap filling process, and the increases of current causes a gradual filling of the states that represents the rise of the activation energy. After the filling of all interfacial states, the charges are injected over the interfacial barrier, which stands for second charge relaxation [53].

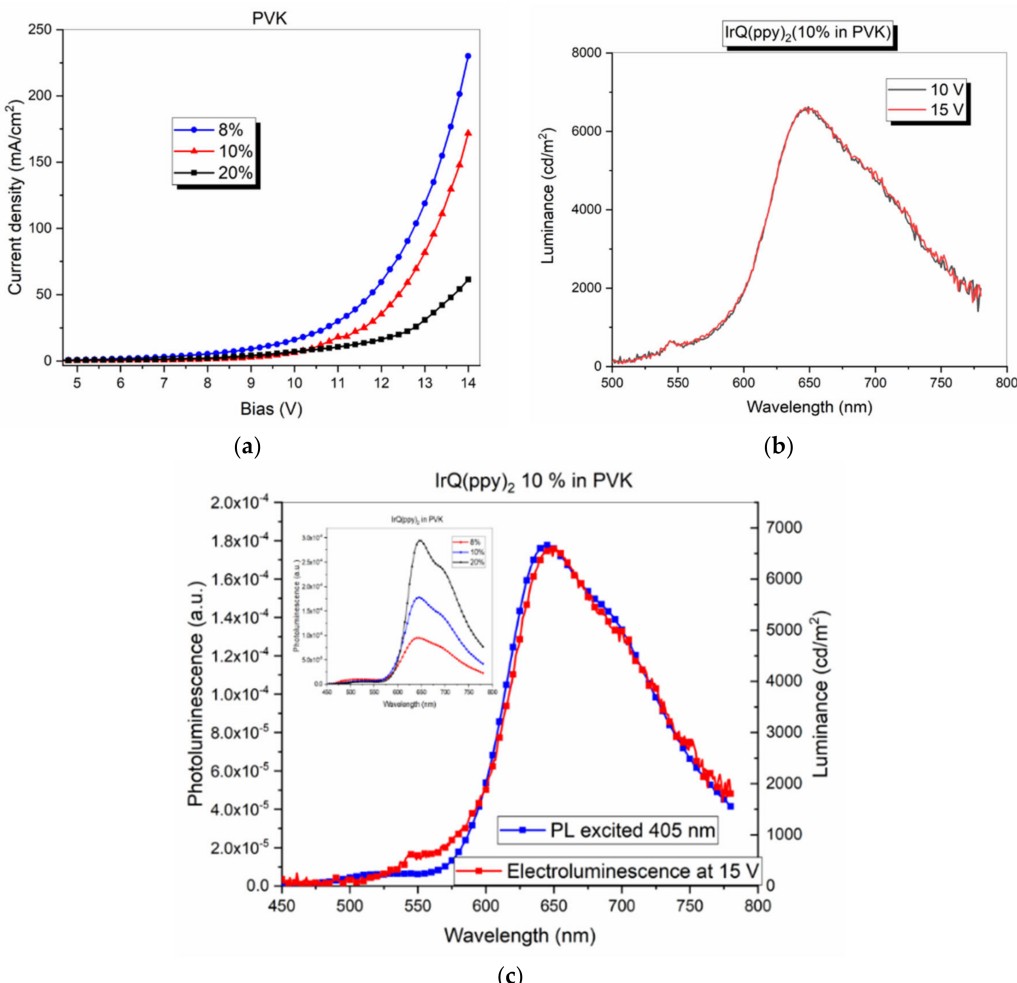

**Figure 3.** (**a**). Current–voltage of IrQ(ppy)$_2$ based OLED in PVK versus concentrations; (**b**) luminance versus applied bias for the 10% IrQ(ppy)$_2$ in PVK; (**c**) luminance and photoluminescence of the IrQ(ppy)$_2$ (10%) in PVK.

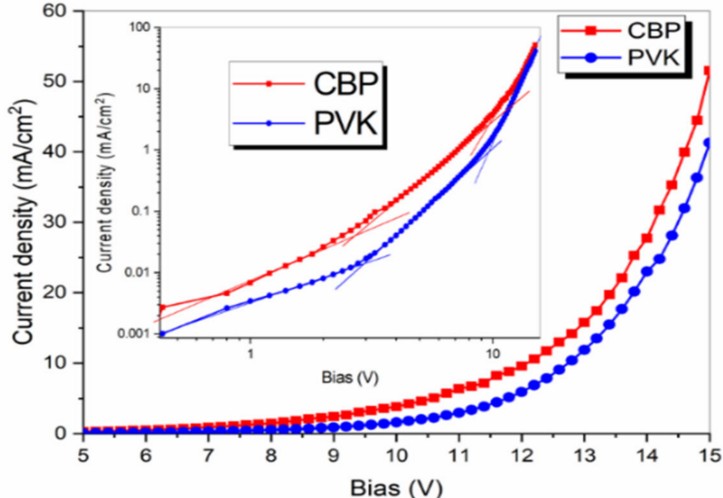

**Figure 4.** Current–voltage of IrQ(ppy)$_2$ based OLED in PVK and CBP (20%). Inlet, log–log *I–V* curves.

### 3.2. Current–Voltage Characteristics: Electroluminescence of IrQ(ppy)$_2$ in CBP

Both types of OLED, with PVK and CBP, present a combination of the red and green electroluminescence, which comes from the two types of ligands. Figure 5a reveals the current–voltage

curves at different concentrations of IrQ(ppy)$_2$ in CBP small-molecule host layer, while Figure 5b,c shows the luminance measured at 14 V for the IrQ(ppy)$_2$:CBP (20%) and IrQ(ppy)$_2$:CBP (10%) samples and the photoluminescence on the same sample excited at 405 nm. As can be seen, there is the same overlap between spectra, like in the case of IrQ(ppy)$_2$ in PVK.

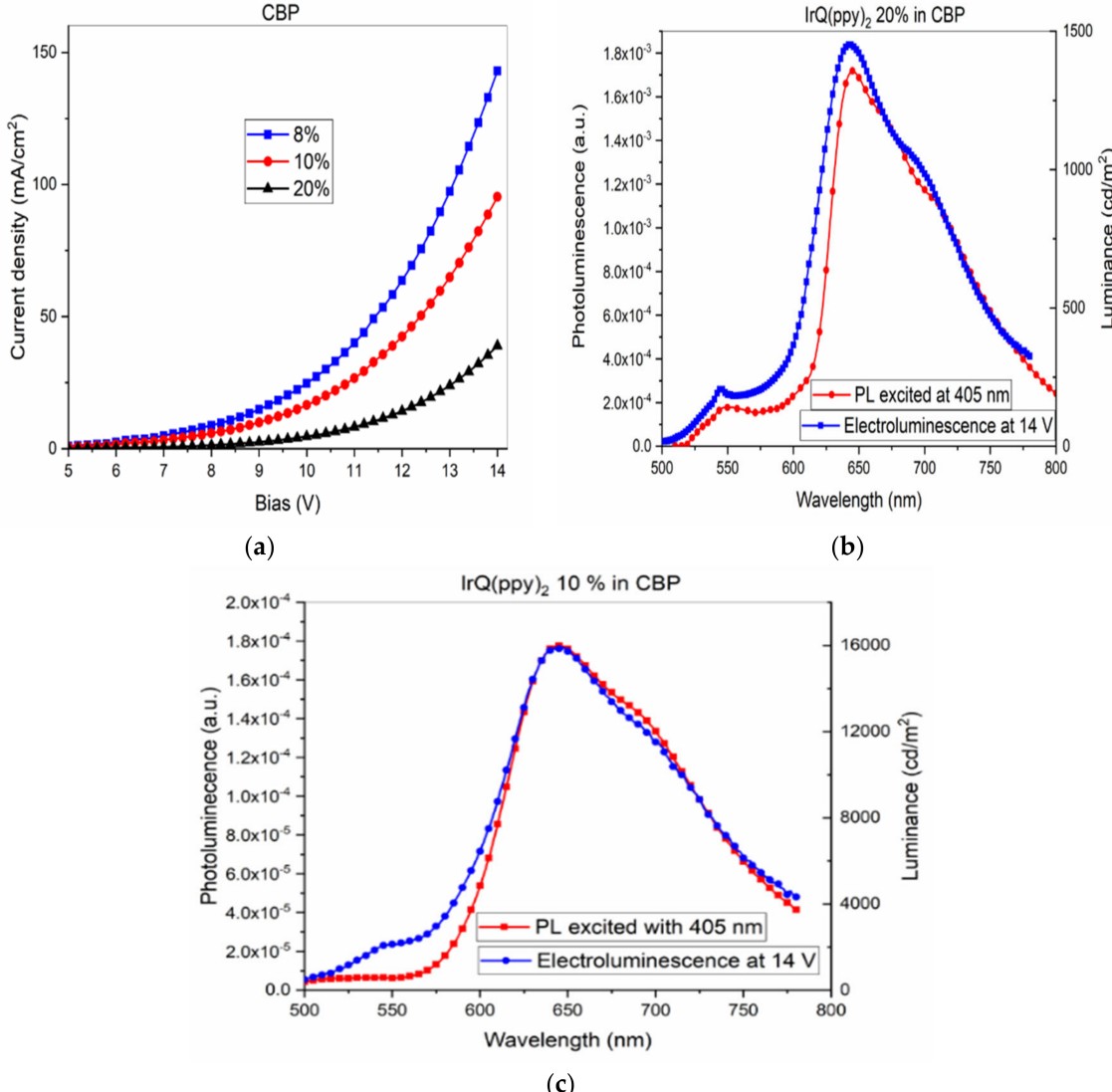

**Figure 5.** (**a**). Current–voltage of IrQ(ppy)$_2$ based OLED in CBP versus concentrations; (**b**) luminance and photoluminescence of the IrQ(ppy)$_2$ (20%) in CBP. (**c**) Luminance and photoluminescence of the IrQ(ppy)$_2$ (10%) in CBP.

The luminance increases with the applied voltage in the case of IrQ(ppy)$_2$ in CBP, giving an orange color as combination between the red and green electroluminescence to just the red one. Figure 6a shows the luminance variation versus the applied voltage. In Figure 6b is given the color variation on the CIE circle and the changing of colors from orange one (590 nm) at 12 V applied voltage to the red color (600 nm) at 15 V applied voltage.

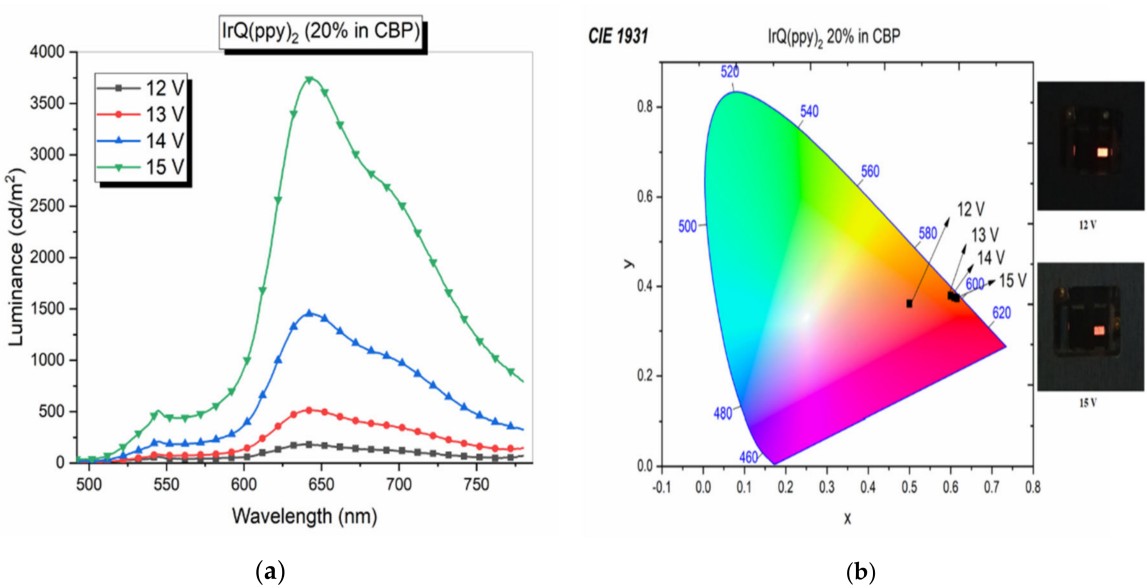

(**a**)　　　　　　　　　　　　　　　　　　　(**b**)

**Figure 6.** Luminance versus bias voltage (**a**); CIE and image colors versus bias of IrQ(ppy)$_2$ in CBP(**b**).

Because the IrQ(ppy)$_2$ in PVK does not change their luminance with the applied voltage, even at higher voltages, up to 16 V, we resume the comparisons only to the devices of IrQ(ppy)$_2$ in CBP (10% and 20%). In Figure 7a,b is given the current density and luminance curve as a function of applied voltage for these two devices.

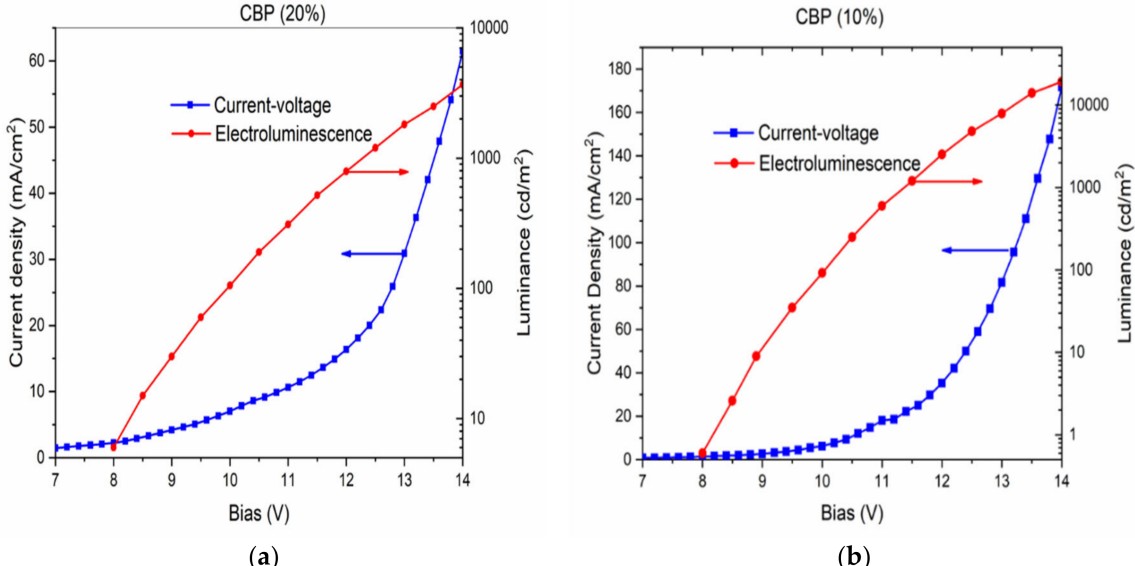

(**a**)　　　　　　　　　　　　　　　　　　　(**b**)

**Figure 7.** (**a**) The current density and luminance curves as a function of applied voltage for IrQ(ppy)$_2$(20%) in CBP. (**b**) The current density and luminance curve as a function of applied voltage for IrQ(ppy)$_2$(10%) in CBP.

For an emitting area of about 4.5 mm$^2$, the current density was 62.6 mA/cm$^2$ at 14 V, in the case of IrQ(ppy)$_2$:CBP (20 wt.%), while the maximum electroluminescence was measured at 3690 cd/m$^2$. The current efficiency is 5.91 cd/A.

To underline the effect of organometallic concentration for the same emitting area, the current density was 172 mA/cm$^2$ at 14 V for the IrQ(ppy)$_2$:CBP (10 wt.%), while the maximum electroluminescence was measured at 19,200 cd/m$^2$. The current efficiency was calculated at 11.2 cd/A.

All parameters are summarized in the following Table 1.

**Table 1.** Electroluminescent efficiency parameters.

| IrQ(ppy)$_2$ | Electroluminescence at 14 V (cd/m$^2$) | Current–Voltage at 14 V (mA/cm$^2$) | PE (lm/W) | CE (cd/A) |
|---|---|---|---|---|
| 10% in CBP | 19,200 | 172 | 2.5 | 11.2 |
| 20% in CBP | 3680 | 62.7 | 1.3 | 5.9 |

## 4. Discussion

The synthesized IrQ(ppy)$_2$ compound with dual emission reveals interesting aspects concerning the electroluminescence properties. Dispersion degree in the conductive thin films significantly influences the electroluminescence, starting with the low concentration (below 10 wt.%) and moving up to the high concentration (over 20 wt.%) in CBP. This fact can be explained by aggregation processes of IrQ(ppy)$_2$ from dispersed organometallic molecules toward aggregated clusters which may induce π–π stacking processes between adjacent molecules. This stacking leads to an enhancement of the red color between the quinoline ligands, depending on the IrQ(ppy)$_2$ concentration in the active layer. The phenomenon is known as inter-ligand energy transfer between two π–π stacked IrQ(ppy)$_2$ molecules.

The use of the LiF buffer layer considerably increases the current injection in the OLED devices, leading to a significant rise in the electroluminescence with low power consumption. This buffer layer reduces the injection barriers between cathodes and the active thin films. In the ohmic zone, the mobile charge carriers are intrinsically present, so that the charge carriers' density needs to be known to say something about the charge mobility. SCLC is applicable to single carrier injection devices, in this case for electron transport through the quinoline ligands of the IrQ(ppy)$_2$, in a similar way with the Alq$_3$ compound used as electron transport material [54]. In our case, the organometallic has both capabilities, acting as ETL and as electroluminescent material.

The overlap between the photoluminescence and electroluminescence spectra reveals the efficiency of the organometallic to produce light emission, with both processes being located on the same molecule. On the contrary case, the electroluminescence is given via an energy transfer between the excitons produced in the conductive thin film and the IrQ(ppy)$_2$. This energy transfer appears in similar systems composed by PVK and doped dye molecules like the Eu TTA phen [55].

Unlike the case of IrQ(ppy)$_2$ in solution, higher red electroluminescence than the green one is observed in the highly doped structures. The explanation is related to the position of the triplet states responsible for the two ligands, quinoline and phenylpiridine. The triplet state responsible for the red emission is located at lower energy than the triplet state responsible for the green emission. In a diluted solution, single molecules are distributed randomly and independently from the other molecules. Usually, a single emission band due to the lowest excited state is observed in organic compounds (Kasha's rule [56,57]). In IrQ(ppy)$_2$, dual electroluminescence is allowed by the nearly perpendicular orientation of the phenylpiridine and quinoline ligands [58]. Unlike the amorphous phase with randomly oriented single molecules, in the aggregated phases, the interaction between the neighboring IrQ(ppy)$_2$ molecules gives rise to the energy transfer from the upper triplet state to the lower triplet state of the neighboring molecules. This is due to the nearly perpendicular orientation between the ppy ligand of an IrQ(ppy)$_2$ molecule and the Q ligand in its neighboring molecule, resulting in enhancement of the red emission and reduction of the green emission by inter-ligands' energy transfer. This energy transfer is propagated through the π–π stacked IrQ(ppy)$_2$ molecules, formed by quinoline connections between adjacent molecules and requires higher energy for propagation. The doping level changes the coherence length defined as the minimum size of aggregate in the out-of-plane direction between molecules [37]. In this context, an increase of the applied voltage is expected for the red electroluminescence, as compared with the green one, together with the decreasing of performances (efficiency) compared with single-color devices based on Ir(ppy)$_3$ or Alq$_3$ electroluminescence.

Concerning the efficiency of these devices, two main parameters can be defined:

- Current efficiency in *cd/A* as the following equation:

$$CE\left(\frac{cd}{A}\right) = \frac{L\left(\frac{cd}{m^2}\right)}{j\left(\frac{mA}{cm^2}\right)} \tag{3}$$

where the *L* (cd/m$^2$) is the electroluminescence and *j* (mA/cm$^2$) the current density.

- Power efficiency in *lm/W* as the following equation:

$$PE\left(\frac{lm}{W}\right) = \frac{L\left(\frac{cd}{m^2}\right) * \textit{device area}\left(m^2\right) * \pi}{j\left(\frac{mA}{cm^2}\right) * V} \tag{4}$$

Similar parameters have been obtained by Y. Sun et al. for the asymmetric configuration of Ir(III) complexes, for which the electroluminescence at 15 V varies between 9713 and 25183 cd/m$^2$. The concentration of Ir(III) complexes varies between 6 and 10 wt.% [59]. Even lower parameters were obtained by H. Yang et al. for the Ir(ppy)$_3$ (8 wt.% in CBP) for the brightness that varies between 2000 and 8000 cd/m$^2$ at 14 V, while the current efficiency is slightly higher from 15 to 18 cd/A at the same bias [31]. More recently, Y. Wu et al. measured the electroluminescence of Ir(piq)$_2$(acac) in different conductive polymers. The obtained values vary between 3819 and 5369 cd/m$^2$ for 11 V applied bias, the current efficiency between 3.96 and 8.35 cd/A and the power efficiency between 1.28 and 3.09 lm/W for 35 nm organometallic neat films. [60]. In the case of Ir(ppy)$_2$(acac), the turn-on voltage is around 2.4 V, while for electroluminescence of 10,000 cd/m$^2$, the power efficiency is about 96 lm/W, which is much better than in our case [61].

From the current–voltage characteristics, the turn-on voltage is around 3 V, which is slightly higher than in other devices. The increasing of the IrQ(ppy)$_2$ concentration was necessary for balancing the orange color electroluminescence of this organometallic because the Ir(ppy)$_2$(acac) is a green single-color OLED. This fact explains the lower performances of the IrQ(ppy)$_2$ OLED.

Compared with the previous results [62], both current efficiency and power efficiency of the OLED devices were improved by adding a LiF buffer layer. This is because, in the previous papers, the aluminum cathodes were deposited directly on the Electron Transport Layer (ETL), drastically decreasing the electron injection in the OLED. The current was 2.23 mA/cm$^2$ for 12 V applied potential in the case of IrQ(ppy)$_2$:TPD (8%) in the previous paper. When an additional LiF layer of around 0.8 nm was stacked between the aluminum cathodes and the ETL layer, the current across the sandwich structure was significantly increased up to 52 mA/cm$^2$ for 14 V applied voltage for the IrQ(ppy)$_2$:CBP (20%) in the present paper (see Figure 4). That means both current efficiency and power efficiency parameters increase by 25 times, with all devices having the same area.

Concerning the choice of the transparent and conductive active layer, significant differences appear in the electroluminescence processes between the PVK polymer and CBP small-molecule material. While the dispersion of IrQ(ppy)$_2$ does not change the electroluminescence versus applied bias in PVK, in CBP, the electroluminescence increases with the applied bias.

## 5. Conclusions

The organometallic compounds dispersed in conductive polymers and deposited as coating thin films can be successfully used for the Organic Light-Emitting Diodes (OLED). The IrQ(ppy)$_2$ organometallic, as an emitting material, has the advantages of dual phosphorescence, red and green, which gives the combined orange-colored electroluminescence and can be successfully used in the RGB displays. Compared with the single-color electroluminescent organometallics like Ir(ppy)$_3$, the efficiency of the IrQ(ppy)$_2$ is quite small, but it has stable electroluminescence. More than that, the green/red ratio can be tuned by varying the concentration of IrQ(ppy)$_2$ or changing the applied potential.

Controlling the doping level and the aggregation processes that can lead to the π–π stackings between adjacent organometallic molecules influences the efficiency of the light emission. Efficient charge transport within the molecular aggregates is one of the main factors which affects the OLED performance. The level of doping with IrQ(ppy)$_2$ in the active layers induces aggregation processes in which the orientation of the quinoline ligands leads to intermolecular transfer across the π–π stacking between molecules and, hence, the energy transfer between molecules that change the electroluminescence properties of the OLED. This fact requires a higher bias for electroluminescence activation of the red emission and, subsequently, the reduction of the electroluminescence, current efficiency and power efficiency of these OLED devices. The choice of additional layers, like conductive polymer as host, can also significantly influence the performances of the OLED based on IrQ(ppy)$_2$ organometallic compound. The use of the CBP small-molecule layer can lead to an increase in the electroluminescence versus the applied voltage. By changing the CBP with a conductive polymer like PVK, the electroluminescence does not vary with the applied bias, suggesting different mechanisms of aggregation in these active layers.

The charge transport across the amorphous/crystalline interfaces was theoretically studied very recently in metal–oxide frameworks without experimental support [39]. Our results represent a step ahead concerning the problem of electrical conductivity for the OLED. In this context, the intermolecular π–π stacking interactions have major influences on the electrical conductivity of electroluminescent devices.

**Author Contributions:** Conceptualization, S.P.; methodology and final draft, I.C.C.; software, S.P.; investigation, I.C.C. and C.C.C.; writing—original draft preparation, S.P.; writing—S.P. and C.C.C. All authors have read and agreed to the published version of the manuscript.

**Funding:** This work was supported by a grant from the Romanian Ministry of Research and Innovation 12PFE/2018 and Core Program PN19-03 (contract No. 21 N/08.02.2019).

**Acknowledgments:** The authors are gratefully to Eng. Serpil Tekoglu, the Institute of Physical Chemistry and Linz Institute of Organic Solar Cells-Johanness Kepler University Linz for their kind technical support in production of OLED sandwich structures.

**Conflicts of Interest:** The authors declare no conflict of interest.

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
