# Peer review of "Organometallic Coatings for Electroluminescence Applications"

_coatings, doi:10.3390/coatings10030277_

Round 1
Reviewer 1 Report
The resubmitted version is readable and somehow improved compared the first version. In general, this manuscript just studied the role of the concentration of IrQ(ppy)2 in the conductive polymer matrices for the application of OLED. However, the study was not well performed and most of the statements in this manuscript lack the support of experimental results. Thus, in my opinion, the manuscript cannot be accept unless significant improvement can be provided to enhance the quality of the manuscript.
The authors claim the “new” compounds IrQ(ppy)2, but as I mentioned in before, this has been reported by the same group before. In the introduction part, the authors should specify the previous achievements with this molecule in OLED, and the improvements or other findings in this work. This manuscript describes the EL improvement with the LiF buffer layer and conductive polymers or small molecule layers. However, the reference result without LiF buffer layer is missing. The detailed comparison of C-V data, PL, EL spectra of IrQ(ppy)2 in the polymer and small-molecule layer should be provided. The authors try to correlate the concentration of IrQ(ppy)2 to EL performance. However, the PL, EL spectra of different concertation of IrQ(ppy)2 were missing. How is it possible to claim the relationship without any data? It is mentioned the IrQ(ppy)2 shows phosphorescence, again, any data to support it? Lifetime measurement? The manuscript discusses quite a lot to correlate the concentration effect to the molecular ordering in the emissive layer (almost half of the discussion part), however, there is no data to support these statements. Can the authors prove some results of GIXRD or GIWAXS on these coatings? The results part 3.1 and 3.2 can be moved to experimental part, as these just experimental details without any results. For device scheme in Figure 2, can the authors provide the characterization of the thickness of these layers? How are the authors sure the buffer layer is 0.8 nm thick. The discussion of Alq3 in this work is confusion, not clear what the authors want to claim as no results provided here with Alq3. The title is too general and no OPV study was taken in this manuscript
Author Response
Reviewer 1:
The authors claim the “new” compounds IrQ(ppy)2, but as I mentioned before, this has been reported by the same group before.
Answer:
The word “new” was withdrawn from the manuscript, accordingly with the Reviewer’s suggestion.
Concerning the novelty of the present manuscript, the following phase was added in the Introduction part (lines 103-111):
The main purpose of this manuscript is related to the changing of the electroluminescence as a result of the aggregation processes which can lead to the crystalline forms of the organometallic. This fact requires different mechanisms of the charge transport, by hopping in the case of diluted (8%) IrQ(ppy)2 and through the π-π bonding for the partial crystalline sample with 20% IrQ(ppy)2. The variation of the electroluminescence between green and red color was observed for the first time and is correlated with the previous results where the photoluminescence and X-ray diffraction were correlated with the crystallization processes, which take place at higher concentrations of organometallic [38]. Some theoretical predictions were given for metal-oxide frameworks materials concerning the electrical conductivity through π-π stacked molecules [39].
A second phase was introduced in Discussions (lines 344-352):
Compared with the previous results [65], both current efficiency and power efficiency of the OLED devices were improved. This is because, in the previous papers, the aluminum cathodes were deposited directly on the Electron Transport Layer (ETL), which drastically decreases the electron injection in the OLED. The current was 2.23 mA/cm2 for 12 V applied potential in the case of IrQ(ppy)2:TPD (8%) in the previous paper. When an additional LiF layer of around 0.8 nm was stacked between the aluminum cathodes and the ETL layer, the current across the sandwich structure was significantly increased up to 62.7 mA/cm2 for 14 V applied voltage in the case of IrQ(ppy)2:CBP (10%) in the present paper (see figure 4). That means both current efficiency and power efficiency parameters increase by 30 times, all devices having the same area 4.5 mm2.
Some references were added for comparison with the previous works.
Reviewer 1:
However, the PL, EL spectra of different concertation of IrQ(ppy)2 were missing. It is mentioned the IrQ(ppy)2 shows phosphorescence, again, any data to support it? Lifetime measurement? The manuscript discusses quite a lot to correlate the concentration effect to the molecular ordering in the emissive layer (almost half of the discussion part), however, there is no data to support these statements.
Answer:
This paper underlines just the electroluminescence dependence form the IrQ(ppy)2. The photoluminescence data has been published in S. Polosan, C.C. Ciobotaru, I.C. Ciobotaru, T. Tsuboi, ” Crystallization properties of IrQ(ppy)2 organometallic complex films”, J. Mater. Res. 2017, 32, 1735-1740., were in figure 7, is shown the luminance of IrQ(ppy)2 dispersed in CBP (8% and 15 %) under 408 nm excitation which is mainly green at low concentrations and red at high concentration. The same is true for the IrQ(ppy)2 dispersed in dichloromethane (figure 2a) between 0.15 mg/ml to 0.8 mg/ml, but also in PMMA (figure 2 b) between 2.1 wt.% to 14.8 wt. %. The phosphorescence lifetimes were given in the text (lines 118-122).
Reviewer 1:
Can the authors prove some results of GIXRD or GIWAXS on these coatings?
Answer:
We have already measured the GIXRD on Rigaku equipment for some thinner samples and the IrQ(ppy)2 thin film on glass substrates is shown below:
The measurements are still in progress for comparison with the neutron diffraction data.
Reviewer 1:
The results part 3.1 and 3.2 can be moved to the experimental part, as these just experimental details without any results.
Answer:
The chapters 3.1 and 3.2 are denoted as 2.1 and 2.2 and moved in the experimental part accordingly with the Reviewer’s suggestion.
Reviewer 1:
For the device scheme in Figure 2, can the authors provide the characterization of the thickness of these layers?
Answer:
As we mentioned in the Materials and Methods, the thicknesses of the deposited layers were measured with an FR-portable Thetametrisis ellipsometer in the reflection mode, which allows the estimation of the thicknesses up to 10 layers, based on their dielectric constants variations with the reflected wavelength. A graph of just the organometallic-deposited thin film is given below:
Reviewer 1:
How are the authors sure the buffer layer is 0.8 nm thick?
Answer:
The thickness of the LiF layer was determined during thermal evaporation deposition and was given by the quartz balance.
Reviewer 1:
The discussion of Alq3 in this work is confusion, not clear what the authors want to claim as no results provided here with Alq3. The title is too general and no OPV study was taken in this manuscript.
Answer:
All parts of the manuscript and from the title concerning the usage of Alq3 in photovoltaic devices were removed accordingly with the Reviewer’s suggestions.

Reviewer 2 Report
I am rejecting this work on the following basis:
[1] Reported devices displayed maximum brightness lower than 1cd/cm2, i.e. lowest brightness emitted by any light emitting diode. It seems devices are not giving any light.
[2] Operating voltage of reported devices are higher than 14 V that is very high. OLED devices with such operating voltage are useless.
[3] Page-7, line 274, "The current efficiency is 59.1 cd/A." What is meaning of writing this line? If authors are taking reference of any previous work please cite the paper.
If it's the current efficiency of reported device, then authors please explain ''how is it possible that any OLED device are giving maximum brightness lower than 1 cd/cm2, power efficiency 0.1 lm/W but current efficiency ~59 cd/A?"
[4] It seems a report not an original research articles.
Author Response
Reviewer 2:
Reported devices displayed maximum brightness lower than 1cd/cm2, i.e. lowest brightness emitted by any light-emitting diode. It seems devices are not giving any light.
Answer:
The confusion started with the units of brightness which were given in cd/cm2 instead of cd/m2. With this correction, the electroluminescence varies between 3680 cd/m2 for the highly doped sample (20wt.% IrQ(ppy)2 in CBP) to 19200 cd/m2 for the 10wt.% IrQ(ppy)2. Some correction has been done for the current efficiency 5.91 cd/A (and not 59.1 cd/A) and the respective power efficiency of 1.3 lm/W instead of 0.13 lm/W.
Two new figures were added that shows the L-I-V curves for the 10wt.% and 20wt.% IrQ(ppy)2 in CBP to underline the effect of organometallic concentration over the electroluminescent performances of these devices.
Reviewer 2:
Operating voltage of reported devices are higher than 14 V that is very high. OLED devices with such operating voltage are useless.
Answer:
For all references given in the paper, concerning the electroluminescent parameters, the operating voltages vary between 11 to 20 V and the brightness is given for the maxima value. Indeed, the functional point for operating voltage is chosen at a lower value for the stability of these devices.
Reviewer 2:
Page-7, line 274, "The current efficiency is 59.1 cd/A." What is meaning of writing this line? If authors are taking reference of any previous work please cite the paper. If it's the current efficiency of reported device, then authors please explain ''how is it possible that any OLED device are giving maximum brightness lower than 1 cd/cm2, power efficiency 0.1 lm/W but current efficiency ~59 cd/A?".
Answer:
As we stated before, all these parameters were corrected for the samples with IrQ(ppy)2 in CBP.
Reviewer 2:
It seems a report not an original research articles.
Answer:
All the obtained devices were made in the Institute of Physical Chemistry and Linz Institute of Organic Solar Cells-Johanness Kepler University Linz for which we are grateful for the technical assistance a few months ago.

Reviewer 3 Report
The manuscript deals with the interesting subject of the evaluation of organometallic coatings.
In the title it is mentioned photovoltaic applications, nonetheless, there is no photovoltaic application in the main text. In the introduction part, it is necessary to mention precisely the advantage and/or novelty of this work in comparison to the recently published works Refs 39, 46, 47, 49, 50, 56 and 58. In the last paragraph of the introduction the highlights of this work are not obvious. It is also stated that “Moreover, the thickness of the active layer was optimized to increase the OLED performances considering the interplay between amorphous and crystalline structures versus concentration, which can influence the charge transport across the sandwich structure and then further the external quantum efficiency.” but there are not any experimental data presented and discussed in the respective sections. In the schematic representation of the OLED structure in Fig. 1 the electron transport layer is missing and it is not in agreement with the relative description in the text. In the Section 3.3:
- A reference for the Equation (1) has to be cited.
- The authors state that “the increasing of IrQ(ppy)2 concentration reduces the current through the OLED, suggesting a better conversion of the electrical energy into non-thermal emitted light, by electroluminescent processes”. This has to be verified by the presentation of the respective Luminance results that were obtained for all concentrations and not only that derived for the 20% (Fig. 3 b).
- Explanations about the conduction mechanisms for each type of conductive host material are very speculative. In the cited Ref. 51 is stated that “the 2C-SCLC model is fully valid if no injection barrier is present at metal–organic semiconductor interface”. Thus, in order to distinguish between space-charge limited current and interface limited current more deep analysis is required.
- Finally, it is not clear why the authors chose to present the analysis of the J-V characteristics curves only for the samples with 10% IrQ(ppy)2. Especially, if we take into account that the 20% IrQ(ppy)2 in PVK shows the best performance.
In the Section 3.4: “Electroluminescence and luminance measurements”, only selected Electroluminescence, Photoluminescence and Chromaticity results are presented. Thus, what is the point of the first paragraph of this section? Furthermore, why the comparative results of the studied samples and OLED devices are not included? The last two paragraphs in the Discussion section cannot be justified with experimental data and results. So, what is their meaning? Based on the aforementioned comments it is very difficult to follow the Discussion and Conclusions of the manuscript, which have to be rewritten after the suggested revisions of the experimental results.
Thus, I would advise to the authors to add all the appropriate experimental results, reorganize their presentation and modify the manuscript properly in order to proceed further.

Author Response
Reviewer 3:
In the title it is mentioned photovoltaic applications, nonetheless, there is no photovoltaic application in the main text. In the introduction part, it is necessary to mention precisely the advantage and/or novelty of this work in comparison to the recently published works Refs 39, 46, 47, 49, 50, 56 and 58.
Answer:
In all mentioned references, the effect of concentration was monitored only for the photophysical properties of IrQ(ppy)2 in CH2Cl2, PMMA or even CBP (TPD) but only concerning the photoluminescence variations with the doping level and the lifetimes, but not the electroluminescence performances. This is the main purpose of the present paper.
Reviewer 3:
It is also stated that “Moreover, the thickness of the active layer was optimized to increase the OLED
performances considering the interplay between amorphous and crystalline structures versus concentration, which can influence the charge transport across the sandwich structure and then further the external quantum efficiency.” but there are not any experimental data presented and discussed in the respective sections.
Answer:
The phrase was corrected as follow:
Moreover, the IrQ(ppy)2 concentration in the active layer was changed to monitor the OLED performances considering the doping level, which can influence the charge transport across the sandwich structure and then further the external quantum efficiency [44].
And generally, we have removed the crystalline word from the text for future investigation of crystallinity.
Reviewer 3:
The authors state that “the increasing of IrQ(ppy)2 concentration reduces the current through the OLED, suggesting a better conversion of the electrical energy into non-thermal emitted light, by electroluminescent processes”. This has to be verified by the presentation of the respective Luminance results that were obtained for all concentrations and not only that derived for the 20% (Fig. 3 b).
Answer:
A long comparison has been done between 10wt. % and 20wt.% IrQ(ppy)2 devices. Because the samples with PVK do not show any variation of the electroluminescence with the applied bias, the discussion was reduced for the samples in CBP.
The above phrase was re-write as follow:
“the increase of IrQ(ppy)2 concentration reduces the current through the OLED, which influences the electroluminescent processes”.
Reviewer 3:
Finally, it is not clear why the authors chose to present the analysis of the J-V characteristics curves only for the samples with 10% IrQ(ppy)2. Especially, if we take into account that the 20% IrQ(ppy) in PVK shows the best performance.
Answer:
The answer was given above where two concentrations of IrQ(ppy)2 were compared.
Reviewer 3:
Explanations about the conduction mechanisms for each type of conductive host material are very speculative. In
the cited Ref. 51 is stated that “the 2C-SCLC model is fully valid if no injection barrier is present at metal–organic semiconductor interface”. Thus, to distinguish between space-charge limited current and interface limited current more deep analysis is required
Answer:
This aspect is questionable, because many other authors support the idea of the SLSC mechanism when the SCLC is applicable to single carrier injection devices, in this case for electron transport through the quinoline ligands of the IrQ(ppy)2, in a similar way with the Alq3 compound used as electron transport material [52].
This fact was underlined at Discussions (lines 273-276). Probably, the solution is to measure the Hall effect for the charge carriers transport.
Reviewer 3:
The last two paragraphs in the Discussion section cannot be justified with experimental data and results. So, what is their meaning? Based on the aforementioned comments it is very difficult to follow the Discussion and Conclusions of the manuscript, which have to be rewritten after the suggested revisions of the experimental results.
Answer:
Indeed, the part of the manuscript concerning the photovoltaic properties of Alq3 is hard to be followed without experimental details and was removed from the text and title.
Figure 1 was changed accordingly with the reviewer’s suggestion.

Round 2
Reviewer 1 Report
I am OK with the revised version, it can be accepted as it is.
Author Response
Many thanks for the revision.
Reviewer 3 Report
In general, the presented experimental results cannot support the discussion, the arguments and the conclusions of the authors because there is a lack of comparative results. Moreover, the presentation of the results is poor with serious mistakes and mismatches. For example there are mentioned some of them in the following:
- In Figure 3 b) the right axis refers to electroluminescence (legend in figure and axis title) or to luminance (figure caption)?
- Figure 5 shows results for IrQ(ppy)2 in PVK or in CBP?
- What is the scale and the corresponding units of the right axes in Figs. 7 a) and b)?
- On page 7 the authors claim that "The electroluminescence of IrQ(ppy)2 embedded in PVK polymer does not vary with the applied voltage giving the same orange color as a mixture of green and red electroluminescence" and "The electroluminescence intensity increases with the applied voltage in the case of IrQ(ppy)2 in CBP going from a combined orange color between the red and green electroluminescence to just the red color". However, the electroluminescence spectra presented in Figs. 5 and 6 for IrQ(ppy)2 in PVK and in CBP, respectively looks very similar and with the same characteristics.
- In the Figures 4 and 7, Lines 256-262, Table I and Lines 332-335 the results and the discussion don't match.
- Lines 324-325 "From the current-voltage characteristics, the turn-on voltage is around 3V, which is slightly higher than in other devices". Is this a result of the present work? This comment confuses the reader because it isn't in agreement with the presenting experimental results (Figs. 3, 4 and 7) and on the other hand no reference is cited.
Finally, even in the revised version it is not clear what is the scope of the work and what it is concluded by the derived results and their analysis. For all the above I believe that the paper does not meet the quality standards for being accepted for publication in the Coatings.

Author Response
We would like to thank you also for the comments of Reviewer no 3. All of these have important guiding significance for our research work. We took into account all points and the answers are detailed below.
As a general-purpose, there are three main aspects which are followed in this paper:
- The changing of the electroluminescence with the increasing of IrQ(ppy)2 concentration in the active layer. This fact was proven by X-ray diffraction and photoluminescence in the referent 38 and now, was compared with the electroluminescence measurements in CBP.
- The choice of the host conductive material plays a significant role in the electroluminescence results. The embedding of the organometallic in PVK does not change the electroluminescence versus applied voltage, while in CBP small molecule, the electroluminescence increases. This fact suggests different dispersion degrees between these two host materials.
- The adding of LiF buffer layer under aluminum cathodes reduces the injection barrier of electrons in the active layer, improving the electroluminescent parameters of the OLED devices.
The first aspect was described in the Abstract (lines 10-13), in Introduction (lines 141-146) and Conclusions (lines 378-386) with an extension in the Discussions chapter (lines 314-333)
The second one, concerning the choice of materials for the host of the active layers, appears also in the Abstract (lines 17-19), in Introduction (lines 138-141) but also in the Conclusions (lines 388-391).
As for the third aspect, the influence of the LiF layer is described in the Abstract (lines 13-15), in Introduction (lines 144-147) and in Discussions (lines 356-364).
The Results were divided for each host material for a better understanding.
Reviewer 3.
Point 1
In general, the presented experimental results cannot support the discussion, the arguments and the conclusions of the authors because there is a lack of comparative results.
Answer
In the revised manuscript we have added comparative electroluminescent parameters from papers published in 2019 and 2020 (references 61-63) It was added in the text, according to the suggestions of the reviewer, starting with line 331, comparative results with other similar configurations of Ir(III) which gives the same electroluminescence parameters.
Moreover, the results from this paper were compared with a similar organometallic compound in CBP (8%) for which the electroluminescence varies between 2000 cd/m2 to 8000 cd/m2, (ref 62), while in our case varies between 3900 cd/m2 for the sample with 20% IrQ(ppy)2 in CBP to 19200 cd/m2 for 10% IrQ(ppy)2 in the same CBP.
Another comparison was discussed based on the paper published in Organic Electronics (ref 63 from 2020) for an asymmetric Ir(piq)2(acac): the electroluminescent parameters are even lower between 3812 cd/m2 to 5369 cd/m2. But in this case, the electroluminescence gives mainly the red color, while in our case is a mixture of the green and red electroluminescence.
Point 2
Moreover, the presentation of the results is poor with serious mistakes and mismatches. For example there are mentioned some of them in the following:
- In Figure 3 b) the right axis refers to electroluminescence (legend in figure and axis title) or to luminance (figure caption)?
The main misunderstanding comes from the assignment of the electroluminescence as luminance or brightness, which is commonly used in the literature. However, to avoid this misunderstanding we have corrected the luminance in electroluminescence in the text but also in the figures and figure captions.
Concerning the figure 3 b) this one was replaced with the electroluminescence of IrQ(ppy)2 in PVK.
Point 3
- Figure 5 shows results for IrQ(ppy)2 in PVK or in CBP?
Answer
The electroluminescence and photoluminescence (figure 5 b) refer to the sample with IrQ(ppy)2 20% in CBP and we have made the following changes:
- We add the name of the sample in figure 5 as the title “IrQ(ppy)2 20% in CBP”.
- The caption was completed as following “Electroluminescence and photoluminescence of the IrQ(ppy)2 (20%) in CBP.”
Figure 5 a), describe the current-voltage curves versus IrQ(ppy)2 in CBP, for comparison with the curves in PVK.
Point 4
- What is the scale and the corresponding units of the right axes in Figs. 7 a) and b)?
Answer
The Oy scales in figures 7 a and b are given in candela per square meter, but these values of electroluminescence are represented in the logarithmic scale to be compared with the current-voltages measurements.
Point 5
- On page 7 the authors claim that "The electroluminescence of IrQ(ppy)2 embedded in PVK polymer does not vary with the applied voltage giving the same orange color as a mixture of green and red electroluminescence" and "The electroluminescence intensity increases with the applied voltage in the case of IrQ(ppy)2 in CBP going from a combined orange color between the red and green electroluminescence to just the red color". However, the electroluminescence spectra presented in Figs. 5 and 6 for IrQ(ppy)2 in PVK and in CBP, respectively looks very similar and with the same characteristics.
Answer
To clarify this aspect, we have added a new figure 3b) in which the electroluminescence of IrQ(ppy)2 (10%) in PVK is compared for 10 and 15 V applied voltages. As can be seen, the electroluminescence does not vary with the applied voltages.
Point 6
- In the Figures 4 and 7, Lines 256-262, Table I and Lines 332-335 the results and the discussion do not match.
Answer
Figure 4 gives the current-voltage curves for the samples with IrQ(ppy)2 20% in PVK and CBP and, in the inlet, the same curves are given in the logarithmic scale to identify the mechanisms of the electrical conductivity.
But, as we stated in the previous answer, the electroluminescence values do not vary with the applied bias for the samples in PVK even if the current is slightly higher in these samples. We restrain the later discussions from the text to those in CBP as follow:
- In figures 7a and 7b but in the table are given the electroluminescence for the sample in CBP 20% and 10%.
- In the table, we are given the best performances for the same two concentrations of IrQ(ppy)2 in CBP, together with the current efficiency and power efficiency and these results are compared with some recent publications as it was underlined above.
Point 7
- Lines 324-325 "From the current-voltage characteristics, the turn-on voltage is around 3V, which is slightly higher than in other devices". Is this a result of the present work? This comment confuses the reader because it isn't in agreement with the presenting experimental results (Figs. 3, 4 and 7) and on the other hand no reference is cited.
Answer
This conclusion was extracted from the inlet of figure 4 were for the sample in CBP the turn-on value is around 2.8 V while in the case of PVK polymer, this value is around 3 V. This result is compared with the value of 2.4 V obtained for an asymmetric Ir(ppy)2(acac) [reference 63]. Based on this analogy, in the log-log representation for the current-voltage curves for the concentration IrQ(ppy)2 in PVK, this turn-on value varies between 2.7 to 3 V.
Point 8
Finally, even in the revised version, it is not clear what is the scope of the work and what it is concluded by the derived results and their analysis. For all the above I believe that the paper does not meet the quality standards for being accepted for publication in the Coatings.
Answer
The scope of this paper follows three main aspects that were briefly described below:
- The variation of the electroluminescent parameters with the concentration of dopants. In this sense, three different concentrations from 8%, 10% and 20% wt. IrQ(ppy)2 were detailed in the paper. The increasing concentration leads to aggregation phenomena which completely change the electrical conductivity mechanism between the amorphous or dispersed molecules for 8% wt concentrations to molecular aggregates which can lead to crystallization processes in highly doped samples. While in the first case, the conductivity is given by the hoping mechanism between adjacent molecules, in the highly doped samples, the aggregation properties can lead to crystallization processes through π-π stacking connections between adjacent molecules. This IrQ(ppy)2 was choose due to the presence of quinoline ligands which leads to these π-π stacking connections between adjacent molecules and changes the electrical conductivity along with those connections. This fact was underlined in reference 38 where the diffraction patterns of IrQ(ppy)2 dispersed in CBP change the coherence length between 8% wt to 15%wt IrQ(ppy)2. In this reference [38] was concluded as follow:
“In this case, the orientation of the quinoline ligands leads to intermolecular transfer across the π-π stacking between molecules. According to the Forester energy transfer model, the transfer rate is increased with a decreasing of the distances between neighboring molecules.” [ref. 38].
Our findings from the manuscript submitted to the Coatings journal, give supplementary proves to the above results by including the electroluminescent parameters. The main conclusion is (line 321):
” Unlike the amorphous phase with randomly oriented single molecules, in the aggregated phase, the interaction between the neighboring IrQ(ppy)2 molecules gives rise to the energy transfer from the upper triplet state to the lower triplet state of the neighboring molecules. This is due to the nearly perpendicular orientation between the ppy ligand of an IrQ(ppy)2 molecule and the Q ligand in its neighboring molecule, resulting in enhancement of the red emission and reduction of the green emission by interligands energy transfer. This energy transfer is propagated through the π-π stacked IrQ(ppy)2 molecules, formed by quinoline connections between adjacent molecules and require higher energy for propagation.”
- The changing of the conductive polymers are somehow connected with the above statements. These conductive polymers lead to different luminescence parameters. While the IrQ(ppy)2 in CBP gives different electroluminescence intensities, for different applied voltages, in the case of PVK there are no differences even at higher voltages 16 V. This fact is connected with the π-π stacking process between the quinoline ligands that leads to different aggregation processes in these two conductive matrices.
- The adding of LiF layers reduces the injection barriers for the electrons into the active layer. While in the previous paper [reference 65] the current density was 2 mA/cm2 for 12 V applied bias without the LiF layer, in the present paper the current density was increased to 152 mA/cm2 for 14 V applied voltage in the 8%wt IrQ(ppy)2. In this manner, the current efficiency and power efficiency were increased and consecutively, the electroluminescence was increased.
All of those three aspects were underlined in the abstract, results and especially in the Conclusions chapter.
Hoping that the present corrections clarified the manuscript, we are available for any further comments to change the recommendation for being accepted in this revised form.

Round 3
Reviewer 3 Report
- In general the electroluminescence is a spectroscopic property with arbitrary units whereas the luminance is a photometric measure of the luminous intensity per unit area with cd/m2 units and it can be measured versus the applied bias voltage. Thus, I would advice to the authors to use the terms electroluminescence and luminance accordingly either in the text and the figures.
- The first paragraph of 3.2 section should be moved to the previous section (already presented results for PVK host), and replace the term electroluminescence with the term luminance (see comment #1).
- In Figure 5 b) the electroluminescence for the IrQ(ppy)2 20% in CBP is ~3800 cd/m2 whereas for the same sample in Figure 6 a) is ~1400 cd/m2 for the same bias voltage (14V). Please check the presented results and explain.
- Lines 237-240: "... while the figure 3 b) shows the electroluminescence versus applied bias for the sample with 10% IrQ(ppy)2 in PVK. As one can see, the use of PVK polymer does not change the electroluminescence intensity with the increasing of bias, giving the same orange color as a mixture of green and red electroluminescence". What about the other two concentrations 8% and 20%? I think that it is crucial to present also the respective results for all three concentrations before proceed to the comparisons only for the IrQ(ppy)2:CBP devices. Also, for the sake of completeness the comparison of electroluminescence and photoluminescence versus wavelength should be presented for all studied samples (IrQ(ppy)2:PVK and IrQ(ppy)2:CBP of 8, 10 and 20%). Otherwise it is rather confused the decision of the authors to focus to specific samples and moreover their conclusions seem to be speculative with no clear evidence and proofs.
- Check the value 152 mA/cm2 at 14 V for the IrQ(ppy)2:CBP (10%wt.) with the Figure 7 b), from which is derived a value larger than 160 mA/cm2.
Unfortunately, I have to remain to my previous evaluation that the presented experimental results cannot support the discussion, the arguments and the conclusions of the authors because there is a lack of comparative results between the studied samples in this work and not only with the literature and other published works, which was performed in the last revised version.
Author Response
Response Letter
Dear Reviewers, thank you again for your comments. We have tried to answer step-by-step, as follow:
Reviewer 3:
- In general the electroluminescence is a spectroscopic property with arbitrary units whereas the luminance is a photometric measure of the luminous intensity per unit area with cd/m2 units and it can be measured versus the applied bias voltage. Thus, I would advice to the authors to use the terms electroluminescence and luminance accordingly either in the text and the figures.
Answer:
All figures concerning the electroluminescence measurements were corrected as Luminance in cd/m2.
Reviewer 3:
- The first paragraph of 3.2 section should be moved to the previous section (already presented results for PVK host), and replace the term electroluminescence with the term luminance (see comment #1)
Answer:
The paragraph was moved in 3.1 section between lines 219-222.
Reviewer 3:
- In Figure 5 b) the electroluminescence for the IrQ(ppy)2 20% in CBP is ~3800 cd/m2 whereas for the same sample in Figure 6 a) is ~1400 cd/m2 for the same bias voltage (14V). Please check the presented results and explain.
Answer:
Figure 5 b was corrected with the luminance obtained at 14 V bias.
Reviewer 3:
- Lines 237-240: "... while the figure 3 b) shows the electroluminescence versus applied bias for the sample with 10% IrQ(ppy)2 in PVK. As one can see, the use of PVK polymer does not change the electroluminescence intensity with the increasing of bias, giving the same orange color as a mixture of green and red electroluminescence". What about the other two concentrations 8% and 20%? I think that it is crucial to present also the respective results for all three concentrations before proceed to the comparisons only for the IrQ(ppy)2:CBP devices. Also, for the sake of completeness the comparison of electroluminescence and photoluminescence versus wavelength should be presented for all studied samples (IrQ(ppy)2:PVK and IrQ(ppy)2:CBP of 8, 10 and 20%). Otherwise it is rather confused the decision of the authors to focus to specific samples and moreover their conclusions seem to be speculative with no clear evidence and proofs.
Answer:
We have taken into account the Reviewer’s suggestion and added two new graphs:
Figure 3 c) shows the overlap between the photoluminescence excited with 405 nm in IrQ(ppy)2 (10% in PVK) and the correspondent one of the luminance for 14 V bias. In the inlet is given the photoluminescence dependence versus the concentration of IrQ(ppy)2 in PVK.
Figure 5c) shows the overlap between the photoluminescence excited with 405 nm in IrQ(ppy)2 (10% in PVK) and the correspondent one of the luminance for 14 V bias for comparison with the sample with 20%. IrQ(ppy)2.
Unfortunately, the luminance for other concentrations of IrQ(ppy)2 in PVK (8% and 20%) cannot be measured because the samples undergo a chemical degradation and their luminance becomes unstable and cannot be compared with the previous one.
Reviewer 3:
- Check the value 152 mA/cm2 at 14 V for the IrQ(ppy)2:CBP (10%wt.) with the Figure 7 b), from which is derived a value larger than 160 mA/cm2.
Answer:
Indeed the current density is 172 mA/cm2 like in the first corrected manuscript. We have made the corrections also for the current efficiency in the table.
As one can see some measurements could be made again and complete the manuscript but some cannot be done because of their chemical degradation, all the electroluminescent measurements been made on the fresh samples.

Round 4
Reviewer 3 Report
Accept after minor revision, improve the figures' quality and text editing.
Author Response
Response Letter
Dear Reviewer, thank you very much for your comments. The manuscript entitled “Organometallic coatings for electroluminescence applications”, authors Silviu Polosan, Iulia Corina Ciobotaru, and Claudiu Constantin Ciobotaru has improved accordingly with the reviewers’ suggestions, mainly by improving the quality of the figures but also the main text.
Reviewer 3
Accept after minor revision, improve the figures' quality and text editing.
Answer
All figures were quality improved by using 1026x784 size (pixels) and 300 DPI for the uniformity of the entire manuscript. The manuscript was checked for the English errors and typesetting for an improved final version, without changing the context. All changes are highlighted in red for clearance.
Thank you again.